# *Lavandula × intermedia*—A Bastard Lavender or a Plant of Many Values? Part I. Biology and Chemical Composition of Lavandin

**DOI:** 10.3390/molecules28072943

**Published:** 2023-03-25

**Authors:** Katarzyna Pokajewicz, Marta Czarniecka-Wiera, Agnieszka Krajewska, Ewa Maciejczyk, Piotr P. Wieczorek

**Affiliations:** 1Institute of Chemistry, University of Opole, 45-052 Opole, Poland; 2Institute of Biology, University of Opole, Oleska 22, 45-052 Opole, Poland; 3Institute of Natural Products and Cosmetics, Faculty of Biotechnology and Food Science, Lodz University of Technology, 90-530 Łódź, Poland

**Keywords:** *Lavandula × intermedia*, *Lavandula hybrida*, *Lavandula angustifolia*, cultivars, essential oil, oil yield, chemical composition

## Abstract

This review article is the first in a series that provides an overview of the biology, chemistry, biological effects, and applications of *Lavandula × intermedia* (lavandin, LI). Despite its prevalence in cultivation and on the essential oil market, lavandin has received limited attention from the scientific community. Remarkably more attention is paid to *Lavandula angustifolia* (LA), which is commonly regarded as the superior lavender and has been extensively researched. Our goal is to provide a comprehensive review of LI, as none currently exists, and assess whether its inferior status is merited. In the first part, we outline the biological and chemical characteristics of the plant and compare it to the parent species. The chemical composition of lavandin oil is similar to that of LA but contains more terpenes, giving camphor notes that are less valued in perfumery. Nevertheless, lavandin has some advantages, including a higher essential oil yield, resulting in reduced production cost, and therefore, it is a preferred lavender crop for cultivation.

## 1. Introduction

Lavender has been used for years in traditional medicine and aromatherapy, as well as in the perfume, cosmetic, and food industries. It is also often planted in household gardens and its herb is often used as a fragrance for shelves and wardrobes. The most well-known species of lavender is *Lavandula angustifolia* Miller (LA) and its essential oil (EO) is often portrayed in the popular press and various guides as a panacea—the wonder remedy for multiple conditions. *L. angustifolia* is the only species of the genus Lavandula that is officially recognized as a medicinal plant, described in the European pharmacopeia and used in modern phytotherapy. The common names of LA include common lavender, English lavender, and also true lavender. This species is the most studied and known lavender species, but interestingly, it is not the most cultivated species. The most cultivated is *Lavandula × intermedia* (LI). It is a sterile hybrid of true lavender (*L. angustifolia*) and spike lavender (*L. latifolia*). It is a hardier, larger plant than *L. angustifolia* and yields more plant material per hectare of cultivation. Moreover, *L. × intermedia* contains more essential oil per kg of plant material than true lavender. Therefore, the yields of oil produced per hectare are much higher. In consequence, the obtained EO is much cheaper than true lavender oil and is often used for counterfeiting true lavender oil. The other names for *L. × intermedia* are lavandin, Dutch lavender, or bastard lavender [1,2,3,4,5,6,7].

The common names reflect the common perception of the two plants described above—a bastard lavender as opposed to the superior, the only true, lavender. Is it justified? Certainly yes, if we only consider the perfume industry. Lavandin essential oil is cheaper and has sharper smell notes caused by a higher content of camphor, borneol, and 1,8-cineole [2,7,8,9]. However, is lavandin EO less valuable than true lavender oil in other non-perfumery applications? It might be difficult to answer this question as most of the available research relates to LA. The commercial success of lavandin did not translate into the amount of scientific research on the species. The search phrase “Lavandula and angustifolia” yields 1662 results in Scopus, including 125 review articles (as of October 18, 2022). The phrase “Lavandula and intermedia” renders only 199 articles with nine reviews but not a single review is genuinely dedicated to this plant. Thus, the aim of this article is to provide a comprehensive review of the *Lavandula × intermedia* plant and lavandin EO, its chemical constituents, proven biological activities, and resulting applications in industries and everyday life. In this review, we avoided generalizing properties found and studied in LA to LI, a common practice even in scientific literature. Therefore, the information provided in this article relates to the studies performed or confirmed specifically on *L. × intermedia*. The first, part of the review article addresses the biological and chemical aspects of LI. The second part of the review, which is published separately [10], focuses on the established biological properties and provides an overview of its applications in industries and daily life.

## 2. The Biology of Lavandin

### 2.1. The Taxonomy and Nomenclature of Lavandula × intermedia Emeric ex Loisel 

The genus Lavender—*Lavandula* L. belongs to the family *Lamiaceae* Lindl. According to different classifications of the 20th century, the genus contains from 25 to 39 species. When adding subspecies and intersectional hybrids, the total number of taxa within the genus reaches about 90 [6,11,12,13]. Due to morphological differences and habitat preferences, all taxa are grouped into six sections: *Lavandula* (=section *Spica* Ging.); *Dentatae* Suarez-Cerv. & Seoane-Camba; *Stoechas* Ging.; *Pterostoechas* Ging.; *Subnudae* Chaytor and *Chaetostachys* Benth. *Lavandula × intermedia* Emeric ex Loisel belongs to the section *Lavandula* and is recognized as a spontaneous hybrid of *Lavandula angustifolia* Mill. and *Lavandula latifolia* Medik [12,14]. In the literature, at different periods, *Lavandula × intermedia* is known as *Lavandula hybrida* Reverchon, *L. × hortensis* Hy, *L. × burnati* Briquet, *L. × spica-latifolia* Albert, *L. × aurigerana* Mailho, *L. × senneni* Faucaud, *L. × feraudi* Hy, *L. × guilloni* Hy, *L. × leptostachya* Pau [12,15,16,17,18]. Additionally, recent research presents two infraspecific taxons of *Lavandula × intermedia*: *Lavandula × intermedia nothosubsp. intermedia*, which is typical lavandin infraspecies, and *Lavandula × intermedia nothosubsp. leptostachya* (Pau) Mateo & M.B. Crespo, which is distinguished from the north-eastern part of the Iberian Peninsula [19]. A detailed classification of the taxa is presented in Table 1.

The history of lavender taxonomy dates back to ancient Rome and Greece, where lavender was used as a medicinal and aromatic plant [19. 80]. The first comprehensive monograph of the *Lavandula* genus was published in 1780 by the Linné [20], titled “De Lavandula”, which described six species and eight varieties. This study included LA and LL, but there was no information on LI. The first published data on lavandin appeared in Flora Gallica in 1828, which contained information on the species’ morphology and distribution. The authors classified lavandin as a separate species in the *Lavandula* genus, presented a brief plant description and indicated the Alps Province as the region of occurrence [21]. After this publication, many studies on lavender were published. One of the most important monographs about the genus *Lavandula*, titled “A taxonomic study of the Genus *Lavandula*” [11], presented a classification of 28 species and their varieties arranged in five sections. *Lavandula × intermedia* was recognized as the hybrid of *Lavandula officinalis* Chaix and *Lavandula latifolia* Vill. In turn, *Lavandula angustifolia* was described as the variety of *Lavandula officinalis*. All three taxa were grouped into one section—*Lavandula spica*. Currently, the most up-to-date study presenting the classification of the *Lavandula* genus is “The taxonomy of the genus *Lavandula L*.” by Tim Upson [12]. According to this work, the genus *Lavandula* comprises six sections; *Lavandula × intermedia* and its parent plants are included in the *Lavandula* section.

### 2.2. Geographical Distribution

Lavandula is naturally distributed in the Mediterranean, the Canary Islands, and India. Some species can naturally grow in northern, eastern, and southern Africa, Bulgaria, Spain, Poland, Turkey, France, England, Russia, Australia, and the USA [22,23]. LI, LA, and LL are native to central and southwestern Europe, and originally come from the Mediterranean basin [7,12]. *L. angustifolia* grows naturally in Italy, France, and Spain in mountainous regions between 600 m and 1200 m above sea level. It is a common species in sunny places with well-drained calcareous soils [7,24,25]. *L. latifolia* occurs in the same area but at an altitude of 300 to 800 m. This plant usually grows in mixed forests on limestone-derived soils [25,26]. *L. × intermedia* appears in locations where two parent species are presented simultaneously (France and Spain). Given the habitat requirements of the parent plants, lavandin can be found in full sun on dry, well-drained stony, calcareous soils [12,25]. All three taxa of lavender are widely cultivated around the world. According to Plants of the World Online, published by the Royal Botanic Gardens (Kew), LA was introduced to Austria, Bulgaria, East Aegean Islands, Germany, Crimea, Marianas, New York, Tunisia, Venezuela, Vermont, Western Himalayas, and *LL* was popularized in Portugal, Sicily, and Yugoslavia [27]. The non-native range of lavandin is poorly investigated. However, Jug-Dujaković et al. state that LI was exported to Europe: France, Belgium, and Germany, then to the United States, Japan, South America, and Africa [28]. The map of the native and non-native distribution of three lavender species according to Plants of the World Online is presented in Figure 1.

The analysis of invasive species databases indicates that none of the described lavender taxa has the status of an invasive plant. However, *L. angustifolia* is listed in the NOBANIS—European Network on Invasive Species database, as an introduced species to Austria, Denmark, Germany, Greenland, Norway, and Sweden. In Sweden, LA is recognized as an established species with rare frequency found in disturbed areas, rocks and lava fields, and urban areas. This status means that the species is not a threat to local biodiversity but requires observation [29].

### 2.3. Species Identification

LA, LL, and LI are all perennial herbaceous dwarf shrubs that can grow up to 50 to 150 cm high. Their inflorescence spikes grow above the leaves and can differ in color from lilac and blue to purple. The calyx and corolla of all taxa are tubular, usually ending with five lobes. Although the general appearance of all three taxa is similar, some morphological features can be used to distinguish them (Figure 2). Lavandin differs from true lavender by its larger size and broader leaves. The inflorescence stalk of lavandin is branched, ending with a lax spike, while true lavender has an unbranched stalk with a compact and short spike. It is more challenging to differentiate lavandin and spike lavender. Both taxa are higher than true lavender and have branched inflorescence stalks. However, lavandin has ovate or rhombic bracts and subtending flowers, while spike lavender’s bracts are linear or lanceolate. Finally, the hybrid lavandin is sterile and does not reproduce by seed [7,12,25,30,31,32]. More detailed information about the difference between LA, LL, and LI is presented in Table 2.

In addition to morphological features, anatomical characteristics may help recognize the target lavender taxa. The results of histo-anatomical studies indicate that the structures of the main organs in the *Lavandula* genus have a typical plan for the *Lamiaceae* family, but individual lavender taxa may differ in micromorphological characteristics [33,34,35]. Ștefan et al. [35] compared the anatomical structure of roots, leaves, and stems for *LA* and *LI* cultivars. They found that in the root structure of true lavender, there are more xylem vessels in the central wood cylinder than in lavandin. At the stem level, they noticed that *Lavandula angustifolia* ‘Ellagance’ can be distinguished by its circular stem cross-section; whereas the other cultivars have a square stem cross-section (a characteristic of the *Lamiaceae* family). In terms of the leaf structure, the researchers observed the difference between the morphology, density, and type of trichomes (hairs). Similar to the observations of other researchers, they found two types of trichomes: covering and secretory (Figure 3). Covering trichomes can be branched or unbranched in outline, while secretory trichomes are hairs with a bi-, tetra-, or octocellular gland. Secretory trichomes are further divided into peltate (with multicellular heads) and capitate (with uni- or bicellular head). The glandular cells of secretory trichomes have a large nucleus and dense protoplasm that lacks a large central vacuole. The secretion accumulates in the subcuticular spaces and diffuses outwards when the cuticle is broken [30,33,35,36]. Ștefan et al. [35] observed the absence of peltate trichomes in the superior epidermis of leaves in lavandin cultivars, distinguishing it from true lavender. Differences in the anatomical structure of the leaf between LA and LI cultivars were also observed by Brailko et al. [30]. They noticed more stomata in the lavandin cultivars than in the true lavender and observed more giant epidermis cells in the true lavender cultivars. An unambiguous indication of the anatomical and histological features that distinguish true lavender from lavandin is complicated, mainly because individual cultivars differ significantly. Therefore, more detailed research is required on this topic.

### 2.4. Cultivars and Cultivation

Commercially, *L. × intermedia* is one of the main taxa cultivated for producing EO and for its horticultural importance. Many cultivars of lavandin are created in connection with different requirements; for example, ‘Alba’ was invented for white flowers, ‘Grosso’ for its tolerance to disease dieback, and ‘Hidcote Giant’ for the strongest scent [5,37]. There is no complete list of LI cultivars. In addition, there often needs to be more understanding in identifying the cultivars because a lavender plant can look different depending on the region in which it is grown, soil conditions, or even weather changes [7,38]. We present the most popular varieties in Table 3 based on research and data from various garden collections.

The cultivation of lavender is not demanding. Once established, the plant does not require complicated treatments. The growing place should be sunny with well-drained soil. The plant prefers soil with a neutral or alkaline reaction, although lavandin is also tolerant of slightly acidic soils. Care treatments are limited to occasional watering, fertilizing, and pruning. Lavender should be watered only during prolonged periods of drought. More dangerous for the plant than the lack of water is its overflow, which in turn causes the rotting of the roots. Fertilization is not obligatory, but lavender produces more flowers with more distinct colors after this treatment. Lavender can be pruned to maintain its shape and cause a second bloom. However, this should be performed to a height of about 22 cm immediately after flowering [5,38,44,45]. Lavender cultivation is threatened by numerous diseases and pests that can destroy the crop. The most dangerous or common are:Root rot—this is a disease caused by several pathogens such as *Fusarium* spp., *Phytophthora* spp., *Pythium* spp., and *Rhizoctonia* spp. Its leading cause is moist soil and low temperatures. The effect of the disease is rotting roots, thus slow wilting and the yellowing or browning of the leaves [52].Alfalfa Mosaic Virus—a viral disease probably transmitted by aphids and human hands. The disease manifests as yellow leaves and smaller sizes [53].Xylella—a bacterial disease caused by *Xylella fastidiosa* and transmitted by sap-sucking insects. The disease manifests as stunted growth and leaves that look like they have been burned [54].

### 2.5. Essential Oil Production

The first cultivation of lavender is known from Europe and dates back to 1600 when the plant was grown for the medicinal properties of the EO [5,7]. Large-scale lavandin cultivation began after the First World War. As a hybrid plant, it exhibits some benefits over parent plants—a phenomenon known as hybrid vigor or heterosis. It is sturdier than true lavender, provides higher crops per hectare, and its biomass contains a higher percentage of EO. This difference can be explained by its morphological features—lavandin has a higher density of secretory structures and, thus, a larger storage capacity than true lavender [6,14]. Generally, the productivity of lavandin is higher than true lavender (on average 120 kg of essential oil/ha compared to 40 kg of EO/ha) [14,55]. Overall, the oil production cost of lavandin essential oil is lower than that of true lavender oil. This contributed to the fact that, by 1930, its production in France only reached 100 tons per year, increasing to 1000 tons by 1960, and further rising to 1439 tons in France alone [7,14,56].

Estimating the global production of lavender EOs is challenging. According to Giray’s analysis [3], the global production of true lavender essential oil in 2018 was about 375 metric tons (MT). Bejar [7] estimated that the production of true lavender oil was 750 MT in 2019, while the production of lavandin EO was 2275 MT (total for ‘Grosso’, ‘Abrial’, ‘Sumian’, and ‘Super’). The largest producer of LA EOs is Bulgaria, followed by France and China, but lavender is also grown in England, Russia, former Yugoslavia countries, Australia, the United States, Canada, South Africa, Tanzania, Italy, and Spain [5,7,45]. The leading producer of LI EOs is France (1439 MT in 2016), Spain (90 MT in 2011), and Morocco. Lavandin is extensively cultivated in Italy, the Balkan Peninsula, Australia, and Tasmania. Spike lavender is less prevalent and is mainly produced in Spain (18 MT in 2011) [5,7,57].

Generally, lavender and lavandin essential oil production shows an upward trend. For example, Bejar [7] reports that true lavender EO production increased in France from 35 MT in 2000 to 109 MT in 2016, and lavandin EO production increased from 950 MT to 1439 MT in the same period. Similarly, in Bulgaria, the production of true lavender EO increased about five times between 2011 and 2017. Lavender essential oils are important raw materials in the cosmetic and medical industries and lavender cultivation can be highly profitable. The economic analysis presented by Giray [3] shows that, in 2011, the mean yield from 1 ha of lavender was 1636 kg in Turkey; the production cost for 1 kg of lavender was calculated as USD0.95, and the net profit counted for 1 ha was around USD 895. Similar results were presented for Bosnia and Hercegovina, with a total profit of USD 1237 per hectare from lavender cultivation. According to Persistence Market Research data, the lavender market was valued at USD 76 million in 2016 and is projected to be USD 124.2 million in 2024 and USD201.6 million in 2032 [3,5,7,45,58].

Summarizing the entire chapter, *Lavandula × intermedia* is a spontaneous hybrid of *Lavandula angustifolia* and *Lavandula latifolia*. It can be distinguished from its parent plants by its larger size (to 150 cm), broader leaves with grey tomentose, and branched inflorescence stalk. The flowers are lilac-purple to white, subtending by ovate or rhombic bracts. Lavandin typically grows in full sun on dry, well-drained stony, calcareous soils. The species is native to the Mediterranean region (France and Spain), but it is currently cultivated in many countries. Commercially, *L. × intermedia* is grown to produce essential oils and for its horticultural importance—the most common cultivars are ‘Super’, ‘Grosso’, and ‘Abrial’. France, Spain, and Morocco are the leading producers of LI EOs. The advantageous biological characteristic of lavandin, combined with an essential oil composition that is generally similar to *L. angustifolia,* and its high yield, contributed to the fact that this taxon is the most popular lavender crop, and lavandin EO dominates among the lavender essential oils on the market.

## 3. The Phytochemicals of Lavandin

### 3.1. Phytochemicals of Essential Oil

Plants from the *Lavandula* genus are valuable raw materials for cosmetics and perfumes as well as aromatherapy and household products. Their essential oils are especially valued and most research on lavender phytochemicals is related to this material. Lavender essential oils are usually produced through steam distillation or hydrodistillation of fresh or dried flowering tops gathered during the flowering season. As previously stated, *L. × intermedia* is a particularly important lavender crop for the essential oil industry due to its higher yield as opposed to *L. angustifolia* [14,55]. The EO content in LI plant material has been analyzed multiple times. Various parts of the plants were analyzed, such as fresh and dried flowers, flowering tops, and dried leaves (Table 4). The percentage of oil content in fresh flowers from different world regions ranged from 0.9% to 3.3%. In dried flowers, the oil yield was higher and varied from 3.6% to 9.9%, except for Spain with reported lower yields (1.3% or lower) [2,6,55,59,60,61,62,63]. Kaloustian et al. [63], and Bajalan and Pirbalouti [16], analyzed the yield of essential oil produced from dried leaves and found that this part of the plant contains significantly less oil than the flowering parts (0.4–1.5%). A comparison of the EO content in LI plant material shows that the oil content is significantly higher than in *L. angustifolia*. As reported by Walasek-Janusz et al. [64], the oil yield of lavandin grown in Poland was 4.4–8.1%, whereas the yield of LA was only 3.1–3.6%. 

Even though *L. × intermedia* is native to the Mediterranean area, lavandin is cultivated in many places and oils from different world regions have been obtained and analyzed, e.g., Spain [55,70], France [71,72], Australia [73], Norway [61], Turkey [14,62,69], Greece [60], United Kingdom [74], Italy [75,76], Croatia [59], USA [77], Romania [78] and Iran [16]. Many different cultivars of this plant have already been investigated. However, the cultivars ‘Grosso’, ‘Abrial’, ‘Provence’, and ‘Super’ are the most popular, best-known, and studied. Regardless of the plant cultivar or origin, over the years, more than 100 different volatile organic compounds have been identified in *L. × intermedia* oil [14,16,55,61,69,70,71,72,73,77,79]. 

Lavandin essential oil, similar to the oils from the parent species—LA and LL, is characterized by a high content of polar terpenoids, especially oxygenated monoterpenes. 1,8-Cineole, linalool, camphor, borneol, terpinene-4-ol, linalyl acetate, lavandulol, and lavandulyl acetate are the most characteristic compounds in these oils (the structures are given in Figure 4). Other compounds, such as monoterpene and sesquiterpene hydrocarbons were also present, e.g., α-and β-pinene, myrcene, sabinene, 3-carene, α-terpinene, α-santalene, germacrene D, (E)-β-caryophyllene, and trans-α-bergamotene, but the levels of these compounds were much lower than those of the polar constituents. The most abundant components of LI essential oils are linalool and linalyl acetate, which can occur from a few percent to exceeding 50%, but usually over 20%. The presence of high levels of linalool and linalyl acetate is desirable because it results in a more pleasant floral aroma and desired pharmaceutical quality. Other dominant terpenoids found in lavandin EO include camphor (ranging from 2–33%, but most frequently occurring at levels from 5–10%), borneol (1–26%, typically a few percent), and 1,8-cineole (2–49%, usually a few percent). Additionally, significant volatile compounds in the EO include terpinen-4-ol (0.4–16%) and α-terpineol (2–10%), which typically contribute up to 5% of the EO. LI essential oil also contains characteristic terpenoids: lavandulol and its ester, lavandulyl acetate. These are present in the EO at levels ranging from a fraction of a percent to 3%. As with linalool-linalyl acetate, the presence of lavandulol and lavandulyl acetate is desirable as it gives the oil a beneficial herbal-rosy scent [2,6].

The essential oil composition of plants of the same species varies due to different factors such as the place of cultivation, weather conditions, harvesting time, etc. The plant variety or cultivar is considered one of the most prominent factors in essential oil chemistry [4,23,80]. The chemical composition of EO produced from the most popular lavandin cultivars, such as ‘Super’, ‘Grosso’, and ‘Abrial’, is presented in Table 5, Table 6 and Table 7, respectively. Researchers investigated oils from different plant raw materials, such as fresh or dried flowers, flowering tops, or even dried leaves. Constituents with content above 3% are bolded in all of the tables concerning the composition of the essential oil.

The qualitative composition of lavandin essential oil from different cultivars is comparable. Across all EOs analyzed, the same group of oxygenated compounds was dominant. The slight differences concerned the content of individual components. The EO of ‘Super’ (Table 5) revealed a higher content of 1,8-cineole (2.6–15.9%) and borneol (1.3–4.2%) than the ‘Grosso’ and ‘Abrial’ cultivars (Table 6 and Table 7) [55,61,71,72,73]. The ‘Super’ essential oil was also generally recognized as containing a high linalyl acetate percentage (35–37%) among the other cultivars, which also results in a scent that is more similar to that of *L. angustifolia* essential oil [6]. It should be noted that this is not an absolute principle, as the examination of Table 5 with examples of ‘Super’ EO compositions illustrates; this varies significantly depending on the location of cultivation. 

The essential oils produced from the ‘Grosso’ cultivar (Table 6) are relatively richer in linalool (22.5–51.3%), terpinen-4-ol (1.5–5.3%), camphor (6.0–12.2%), and lavandulyl acetate (1.5–3.5%) [55,61,71,72,73]. Table 7 presents the qualitative and quantitative composition of the ‘Abrial’ cultivar, which revealed the highest content of camphor, relatively, than the other two varieties. The content of the other main ingredients in ‘Abrial’ EO was in the middle range, between the other two cultivars. 

As seen in Table 5, Table 6 and Table 7, individual chemical compositions provided by various authors differ in detail. Some authors provide a more detailed analysis and also present minor components. Other authors do not investigate the content of minor terpenes. Key components are commonly thought to be the main compounds responsible for the EO’s biological effects. However, even minor components can contribute to the activity. They could be neutral or interact with other constituents, to produce either synergistic or antagonistic effects.

The chemical composition of *L. × intermedia* EO is similar in nature to that of *L. angustifolia* and *L. latifolia*. However, some quantitative differences in the concentrations of the key components are observed. Lavandin essential oil contains a similar or slightly lower content of linalool and linalyl acetate than true lavender, but it contains a significantly higher level of 1,8-cineole, camphor, and borneol [2,14,55,60,71,72,73,81]. These monoterpenoids are camphoraceous, resulting in sharper notes in the LI aroma and reduced value in the perfume industry when compared to true lavender oil. On the other hand, an LI oil richer in camphor can be beneficial in aromatherapy. When comparing LI essential oil to the second parent species—LL, it has significantly less 1,8-cineole, less or similar quantities of camphor, and much more linalyl acetate [2,6]. 

True lavender and lavandin essential oils have a similar chemical composition and corresponding aromas. On the other hand, due to the divergent productivity of the plants, they differ significantly in price—lavandin EO is several times cheaper. Therefore, it is often used as a replacement for true lavender oil or even used to adulterate it [2,7,9]. The oil adulterations along with the natural variability of the chemical compositions of the oils pose a significant challenge when performing replicable scientific research on lavender oils, their reliable and safe applications, and the whole essential oil industry. To address these challenges, various industry standards have been established. The most recognized are the standards of the European Pharmacopeia (Ph. Eur.) and the International Organization for Standardization (ISO). Table 8 presents the existing specification for the chromatographic profile of different lavender oils (the ISO ranges are not presented due to copyright issues). As opposed to spike or true lavender oil, lavandin EO is not considered a pharmacopeial material. There is no specification in the Ph. Eur. for the oil of *L. × intermedia* [82]. Inconsistent with the entries in the Ph. Eur., the World Health Organization (WHO) monographs on selected medicinal plants define lavender oil (*aetheroleum lavandulae*) as essential oil obtained by steam distillation from the fresh flowering tops of *L. angustifolia* Mill. or of *L. × intermedia* [83]. In this context, the WHO has a different approach compared to other regulating bodies, as it permits the use of lavandin as a source of lavender oil and establishes one common specification. The ISO 3515:2002 standard specifies more terpenes in the chromatographic profile of LA oils [84]. It also gives different acceptable ranges for components from different oil origins. These ranges vary depending on the origin and sometimes they exclude each other, which suggests that compliance with the standard does not determine the quality of the oil, but regulates and gives guidelines for the producers of the main oil-producing countries. The ISO also established a general norm for *L. latifolia*—ISO 4719:2012 [85]. However, it did not publish any overall norm for *L. × intermedia.* Bastard lavender only received standards for two cultivars: ‘Grosso’ and ‘Abrial’, namely “ISO 8902:2009 Oil of lavandin Grosso (*Lavandula angustifolia* Mill. × *Lavandula latifolia* Medik.), French type” [86], and “ISO 3054:2017 Essential oil of lavandin Abrial (*Lavandula angustifolia* Mill. × *Lavandula latifolia* Medik.), French type” [87]. This is noteworthy, as lavandin EO dominates the lavender oil market. However, the lack of a general norm for lavandin EO is not necessarily detrimental. Despite being authentic, due to natural variability, some EOs obtained from lavender have a substantially different chemical profile and do not meet the specifications set by the standard-setting organizations. This would exclude many new or unique cultivars or new farmers from other than traditional countries and locations.

### 3.2. Phytochemicals of Other Lavandin Products

Essential oils are valued raw materials obtained through the process of steam distillation or hydrodistillation of plant material. The water byproduct of this process, known as hydrolate or hydrosol (H), is also considered a valuable product. From a chemical perspective, hydrolates are volatile organic compounds, mainly polar oxygenated mono- and sesquiterpenes, dispersed in an aqueous phase. They are used as ingredients in the cosmetic and beverage industry [88,89,90,91]. The chemical composition of *L. × intermedia* hydrolates has been investigated multiple times. Table 9 presents the exemplary summarized test results.

Linalool (19.0–68.5%), borneol (1.4–31.8%), 1,8-cineole (t–28.9%), camphor (0.8–17.5%), terpinen-4-ol (2.3–14.0%), and α-terpineol (1.8–9.0%), *cis*- and *trans*-linalool oxide (0.8–6.0% and 0.6–4.2%, respectively) were the major constituents identified in LI hydrolate. These components were also present in corresponding EOs. Quantitative differences in composition between these two products were observed. Lavandin hydrolates were significantly richer in alcohols, e.g., 1,8-cineole, linalool, borneol, terpinen-4-ol; ketones, such as camphor; and oxides, e.g., *cis*- and *trans-*linalool oxides. On the other hand, the content of esters, such as lavandulyl and linalyl acetate, was much lower than in corresponding essential oils, or these two compounds were even absent.

According to the literature, some relationships between essential oil and hydrolate produced in the same distillation process can be observed. When EO contains a lot of polar compounds (e.g., alcohol, diols, esters), the qualitative composition of EO and hydrolates is similar, but they differ in the levels of individual components. However, if the essential oil consists of predominately nonpolar constituents, as in the case of *Pinus silvestris* L., significant qualitative differences between EO and H occur [89,90]. The same relationship was observed for *L. × intermedia* products. As it produces a relatively polar essential oil rich in oxygenated monoterpenes, the hydrolate contains similar constituents to the EO.

Most studies of the composition of lavandin have focused on the essential oil and hydrolate. However, some other secondary metabolites other than terpenoids were also investigated. Dobros et al. [93] determined the presence of phenolic acids (rosmarinic acid, ferulic acid glucoside, caffeic acid, ellagic acid), flavonoids (morin, isoquercitrin, vanillin), and coumarins (herniarin and coumarin) in flower ethanolic macerates. The composition of phenolic acids, glucosides, and flavonoids in lavandin was consistent with the findings of Torras-Calveria et al. [94] who analyzed essential oil distillation waste products. They found other phytochemicals, such as chlorogenic and rosmarinic acids, coumaric acid-*O*-glucoside isomers, and 3,4,5-trihydroxycinnamic acid-*O*-glucoside. The rosmarinic and chlorogenic acid content was 124 and 215 mg/100 g dry matter, respectively. Lavandin was found to have a higher content of coumarin than true lavender. However, *L. angustifolia* flower extracts had higher concentrations of flavonoids and phenolic acids compared to lavandin [93].

## 4. Conclusions

In this review, we present and discuss the biological and chemical aspects of the lavender hybrid—*Lavandula × intermedia*. The title of this article, “*Lavandula × intermedia*—a bastard lavender or a plant of many values?” was intended to draw more attention to this somehow neglected by the research community species of lavender. Despite its prevalence on the market, most studies are conducted on *Lavandula angustifolia*. Such a situation is not beneficial from a pragmatic point of view because most of the lavender essential oil in use is, in fact, lavandin EO. Additionally, true lavender EO is commonly adulterated—usually by its dilution with lavandin EO. The chemical composition of both oils is similar but not identical. Lavandin essential oil contains mainly linalool and linalyl acetate, the same as true lavender oil. Other components are mostly oxygenated monoterpenes and are also comparable with LA, except for 1,8-cineole, camphor, and borneol, which are more abundant in the lavandin EO. This leads to a slightly different, more spicy fragrance to the oil, with more camphorous notes, which reduces the value of lavandin essential oil in the perfume industry. The differences in terpene profiles may also lead to alterations in biological activities. This issue will be analyzed in detail in the second, following part of our article, “*Lavandula × intermedia*—a Bastard Lavender or a Plant of Many values? Part II. Biological Activities and Applications of Lavandin”[10]. Aside from the perfume industry, *L. × intermedia* has evident advantages over its parent species. It is more resilient and produces greater biomass with a higher content of essential oil, resulting in higher yields. Consequently, lavandin is the most cultivated lavender crop and lavandin essential oil is the main lavender essential oil in the market. 

## Figures and Tables

**Figure 1 molecules-28-02943-f001:**
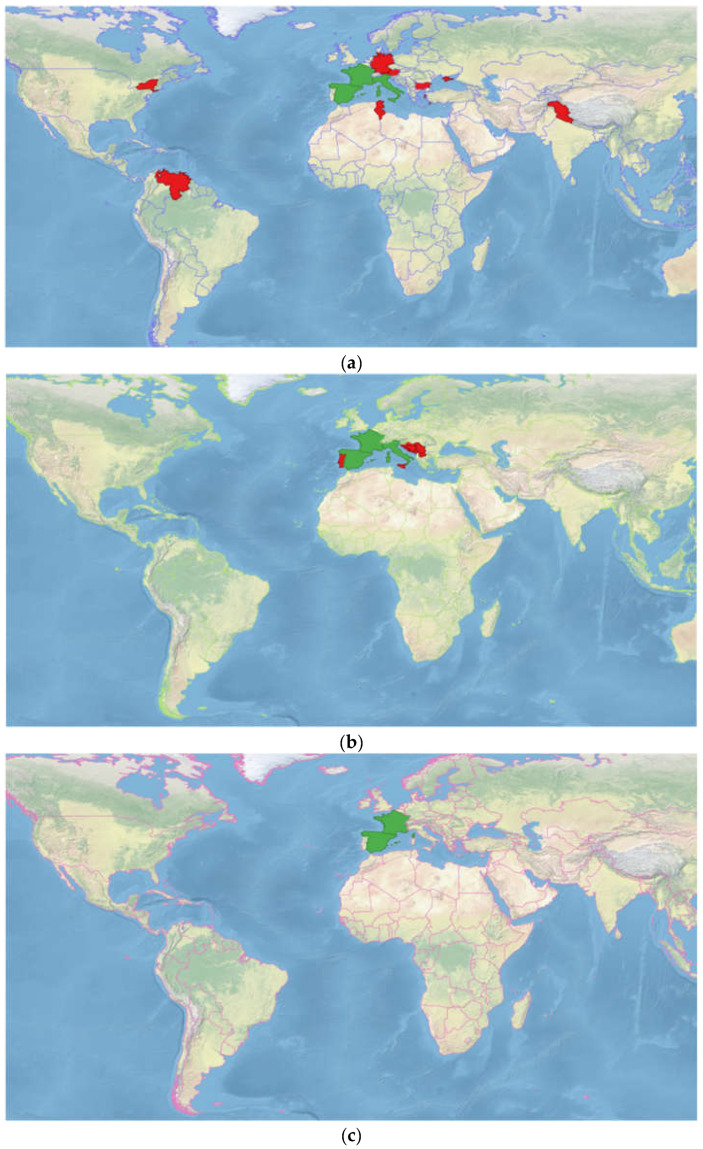
The distribution of (**a**) *Lavandula angustifolia*, (**b**) *Lavandula latifolia*, (**c**) *Lavandula × intermedia* according to *Plants of the World Online*, published by the Royal Botanic Gardens (Kew). The green color represents the native range of distribution and the red color represents the secondary range of distribution. Map source: Made with Natural Earth, https://www.naturalearthdata.com/downloads/50m-natural-earth-1/50m-natural-earth-i-with-shaded-relief-and-water/ (accessed on 2 January 2023).

**Figure 2 molecules-28-02943-f002:**
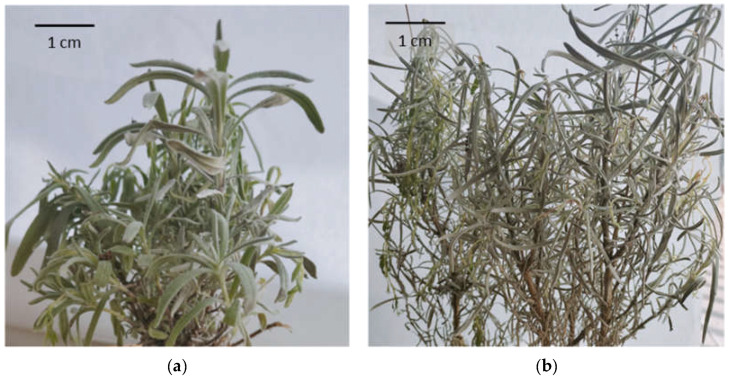
The general appearance of: (**a**) *Lavandula angustifolia* ‘Munstead’; (**b**) *Lavandula × intermedia* ‘Grappenhall’. Photos by M.C.-W.

**Figure 3 molecules-28-02943-f003:**
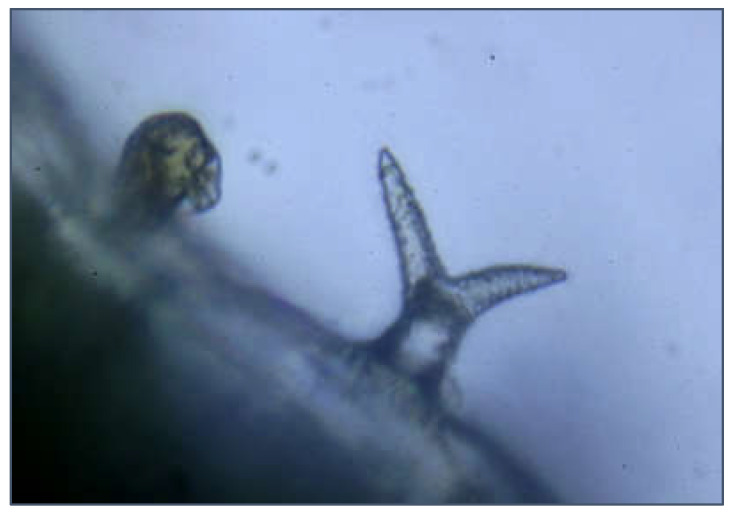
Leaf blade structure in *Lavandula intermedia* ‘Grappenhall’: secretory trichome (on the left), and covering trichome (on the right). Photos by M.C.-W. and Dr. Miłosz Mazur.

**Figure 4 molecules-28-02943-f004:**
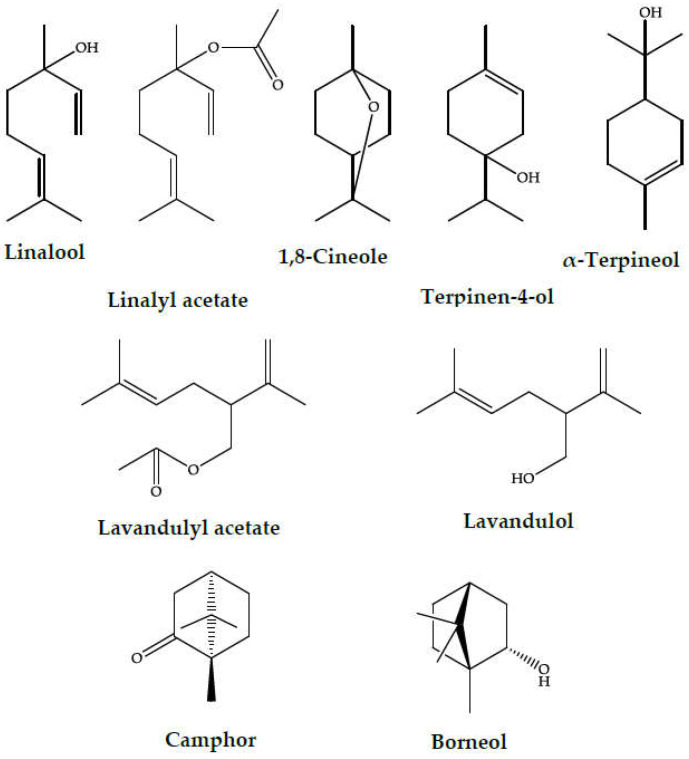
The characteristic monoterpenoids in lavandin essential oil.

**Table 1 molecules-28-02943-t001:** The classification of *Lavandula × intermedia* Emeric ex Loisel [1].

Domain	Latin Name
Kingdom	*Plantae*
Division	*Tracheophyta*
Class	*Magnoliopsida*
Order	*Lamiales* Bromhead
Family	*Lamiaceae* Martinov
Genus	*Lavandula L.*
Section	*Lavandula (=Spica* Ging.)
Species	*Lavandula angustifolia* Mill.
Species	*Lavandula latifolia* Medik.
Hybrids	*Lavandula × intermedia* Emeric ex Loisel
Infraspecies	*Lavandula × intermedia nothosubsp. intermedia*
Infraspecies	*Lavandula × intermedia nothosubsp. leptostachya* (Pau) Mateo & M. B. Crespo

**Table 2 molecules-28-02943-t002:** Comparison of morphological features between *Lavandula angustifolia*, *Lavandula latifolia*, and *Lavandula × intermedia* [7,12,25,30,31,32].

Morphological Feature	*Lavandula angustifolia*	*Lavandula latifolia*	*Lavandula × intermedia*
	Mill	Medik.	Emeric ex Loisel
**Plant habit**	Shrub to 50 cm high	Shrub up to 50–70 (100) cm high	Shrub up to 60–150 cm high
**Leaves**	Shape: narrow, linear-lanceolate	Shape: linear-lanceolate to spathulate	Shape: linear-lanceolate to spathulate
	Color: in younger plants, they are grey tomentose; in older plants, they are green	Color: grey, with silvery-grey indumentum	Color: often grey tomentose
**Inflorescences**	Stalk: unbranched, length approx. 10–25 cm	Stalk: branched, length to approx. 25 cm	Stalk: branched
	Spike: compact in outline, length approx. 4–5(8) cm, sometimes there are small clusters of flowers below the main spike	Spike: trident-shaped in outline, length approx. 5–8 cm, often interrupted	Spike: lax in outline, sometimes interrupted
	Bracts: broad, ovate-rhombic to obovate in outline	Bracts: narrow, linear-lanceolate in outline	Bracts: ovate-rhombic in outline, but varied in shape and size
	Bracteoles: present but minute	Bracteoles: present but small, length to approx. 4 mm	Bracteoles: present but small, length approx. 1–4 mm
**Flowers**	Calyx: thirteen-nerved, has a small circular appendage	Calyx: thirteen-nerved, has a circular appendage	Calyx: thirteen-nerved, has a rotund to elliptic appendage
	Corolla: bilaterally symmetrical, two times longer than calyx, prominent lobes with colorings of blue/mauve, white, infrequently violet to pink	Corolla: bilaterally symmetrical, lobes with colorings of blue to mauve	Corolla: bilaterally symmetrical, lobes are shades of lilac-purple to white
	Flowering time: mid-June to July	Flowering time: from mid-July	Flowering time: from late June to July

**Table 3 molecules-28-02943-t003:** *Lavandula × intermedia* popular cultivar specifications.

Variety	Description	Literature
‘Abrial’	‘Abrial’ was introduced by Professor Claude Abrial in 1920 and was more vigorous and adaptable than the other lavender taxa cultivars. It is an evergreen shrub up to 60 cm high. The flowers have a violet color. The leaves are linear green-grey.	[39]
‘Alba’	‘Alba’ has been known in Europe since 1880. It is an evergreen shrub with a habit of forming a dome up to 75 m high. The name of the cultivar comes from its white flowers collected in long spikes. Its linear leaves have a grey-green color.	[5,8,32,40]
‘Bridget Chloe’	‘Bridget Chloe’ was created in 2014 by John Thomas Hendon from Georgia (US). It is an evergreen shrub up to 75 cm high, with light green to silver-gray leaves. The flowers have dark purple colors. The cultivar is resistant to Lavender Leaf Spot and the Alfalfa Mosaic Virus.	[41]
‘Budrovka’	‘Budrovka’ is a cultivar produced in Croatia and grown for EO. The extract of ‘Budrovka’ has medicinal features—it inhibits bacterial or fungal growth.	[42]
‘Dutch Group’	‘Dutch Group’ is a widely grown cultivar of lavandin introduced in 1920 and historically called ‘Vera.’ Cultivars in the Dutch Group are evergreen shrubs up to 40 cm high; the inflorescence stalk can grow to 91 cm high. Flowers have a dark aster, a violet corolla, and a light green calyx. The leaves are grey.	[5,32,43]
‘Grappenhall’	‘Grappenhall’ is one of the oldest cultivars of lavandin introduced in 1902 and historically called ‘Gigantea’ or ‘Giant Grappenhall’. It is an evergreen shrub up to 36 cm high; the inflorescence stalk can grow to 101 cm high. Flowers have a dark aster, a violet corolla, and a light green calyx. The leaves are broad green-grey.	[5,32]
‘Grosso’	‘Grosso’ is an aromatic French cultivar of lavandin introduced in 1972. It is an evergreen shrub up to 20 cm high; the inflorescence stalk can grow to 76 cm high. Flowers have a violet corolla and a light green calyx. The leaves are narrow, grey-green, and long-stemmed. It is the most commonly cultivated lavender, characterized by many flowers, a high biomass production, and a large amount of extracted EO. This cultivar is resistant to mycoplasmas that threaten the crops of ‘Abrial’.	[5,32,40,44,45]
‘Hidcote Giant’	‘Hidcote Giant’ was introduced in 1958 by L. Johnson. It is an evergreen shrub up to 20 cm high; the inflorescence stalk can grow to 76 cm high. Flowers have a dark aster, a violet corolla, and a light green calyx. The leaves are narrow, grey-green. The advantage of the plant is its stout stems, which is why it is often grown for cut flowers.	[5,32,44,46]
‘Okanagan’	‘Okanagan’ is a cultivar produced for medical reasons. The plant contains an EO enriched with two therapeutic components: 1,8-cineole and borneol, and was used to test the efficacy of these compounds in a murine model of acute colitis.	[47]
‘Old English’	‘Old English’ was introduced in the 1930s and is historically known as *L. spica.* It is an evergreen shrub up to 43 cm high; the inflorescence stalk can grow to 114–117 cm. The flowers are pale-purple. The leaves are narrow, grey-green. This cultivar is ideal for cottage gardens.	[5,32,43,44]
‘Provence’	‘Provence’ has flowers with a dark violet corolla and a light-green calyx.	[32]
‘Riverina Margaret’	‘Riverina Margaret’ was created in 2017 by Nigel Alexander Russell from Australia. It is an evergreen shrub up to 20 cm high with green leaves. Purple flowers form a cylindrical spike.	[48]
‘Seal’	‘Seal’ was introduced in 1955 in England. It is an evergreen shrub up to 33 cm high; the inflorescence stalk can grow to 84 cm high. Flowers have a dark violet corolla and a light green calyx.	[32]
‘Super’	‘Super’ is one of the most common plants cultivated for essential oil with a delightful aroma. It is an evergreen shrub up to 80 cm high. Flowers have a light bluish-purple corolla and a light violet-green calyx. The leaves are narrow green-grey.	[49,50]
‘Sussex’	‘Sussex’ is an evergreen shrub up to 90 cm high with grey-green leaves. This cultivar has the longest blue flowers in the lavender group, making it particularly interesting.	[5,44]
‘Tesseract’	‘Tesseract’ was created in 2019 by Lloyd R. Traven and Richard Grazzini from Pennsylvania. It is an evergreen shrub up to 45 cm high with broad and bright silver leaves. Purple flowers form dense spikes. This cultivar is resistant to leaf spotting and root disease.	[51]

**Table 4 molecules-28-02943-t004:** *L. × intermedia* essential oil yield from different plant materials.

Essential Oil Origin	Fresh	Dried Flowers/	Dried Leaves
	Flowers	Flowering Tops	
	Oil Yield [%]		
Croatia	3.3 [59]		
France		4.5–9.7 [6]	0.4–0.8 [63]
Greece		7.5–8.5 [60]	
Iran			0.5–1.5 [65]
Norway		7.1–9.9 [61]	
Poland		4.4–8.1 [64]	
Romania	2.75 [66]	3.0 [67]	
Spain		0.2–1.3 [55]	
Turkey	0.9–1.7 [68]	3.6–8.4 [68,69]	
Ukraine	0.9–2.0 [2]		

**Table 5 molecules-28-02943-t005:** Exemplary *L. × intermedia* ‘Super’ essential oil compositions from different countries. Values over 3% are bolded (this rule also applies to the following tables concerning chemical composition).

Compound	Greece [60]	Norway [61]	Spain [81]	Spain [55]	Turkey [14]	Turkey [68]
	DF	DF	DF	ND	FF	FF
	Percentage (%)					
α-Thujene	0.1					
Camphene	0.3					
**Octan-3-one**						**3.4**
Sabinene	0.4					1.3
α-Pinene	0.7					1.4
β-Pinene	1.0					
Myrcene	1.0			0.5		1.5
α-Phellandrene	t					
Hexyl acetate	0.2					
3- Carene	0.1					
α-Terpinene	0.1					
***p*-Cymene**	0.1					**5.0**
Limonene			0.7	0.9		
**1,8-Cineole**	**15.9**	**6.8**	**4.3–5.4**	**7.6**	2.6	
(*Z*)-β-Ocimene	2.7		1.7–1.8	0.4		
(*E*)-β-Ocimene	1.9		2.3–2.2	0.4		
γ-Terpinene	0.3					
trans-Sabinene hydrate	0.3					
Terpinolene	0.3					
**Linalool**	**23.0**	**38.5**	**31.4–37.4**	**33.2**	**34.0**	**33.8**
**Camphor**	**11.4**	**3.5**	**5.2–8.6**	**7.1**	**4.8**	**3.8**
Limonene dioxide					0.3	
Linalool oxide					0.4	
Borneol	1.3		2.5–3.1	2.7	4.2	3.4
Lavandulol			0.2–0.9			
**Terpinen-4-ol**	**6.7**		0.4–1.7	**3.3**		0.6
Cryptone	0.3					
Hexyl butyrate	0.7					
**α-Terpineol**	**3.8**			1.5		1.8
*Table continuation…*						
Cumin aldehyde	0.1					
Nerol	0.4					
Hexyl isovalerate	0.1					
Geraniol					0.3	0.8
**Linalyl acetate**	**20.4**	**17.7**	**22.4–28.1**	**29.7**	**47.7**	**34.2**
Bornyl acetate	0.1					
Lavandulyl acetate	0.4	2.2	1.3	2.6		
Neryl acetate	0.7				2.4	2.2
Geranyl acetate	1.3					
(*E*)-β-Farnesene	0.3			1.9	0.5	0.8
Germacrene D	0.3			0.7		
(*E*)-β-Caryophyllene	0.9	1.1		1.4	1.0	
Caryophyllene oxide	0.1				0.3	
α-Santalene	0.1					
α-Bisabolol	1.6			0.4	0.8	
*t*-Cadinol	0.1					

FF—fresh flowers, DF—dried flowers, ND—not determined, t—trace < 0.05%.

**Table 6 molecules-28-02943-t006:** Exemplary *L. × intermedia* ‘Grosso’ essential oil compositions from different countries.

Compound	Australia [73]	France [72]	France [71]	Norway [61]	Spain [55]
	ND	ND	ND	DF	FF
	Percentage (%)				
α-Pinene	1.0	0.6			
β-Pinene	1.1	0.4			
Camphene		0.3			
Myrcene		1.5			0.7–0.8
*p*-Cymene					
Limonene		0.9	0.5–0.6		0.8–1.0
**1,8-Cineole**	**10.9**	**10.2**	**5.4–7.4**	**10.7**	**4.8–6.6**
(*Z*)-β-Ocimene	1.9	1.1	0.8–1.1		0.5–1.3
**(*E*)-β-Ocimene**		0.5	0.3–0.5		**0.4–8.0**
**Linalool**	**34.3**	**22.5**	**28.7–32.1**	**27.9**	**37.7–51.3**
**Camphor**	**7.3**	**12.2**	**6.8–7.7**	**8.1**	**7.7–7.8**
cis-Linalool oxide	0.1				
**Borneol**	**1.6**	**2.9**	**2.1–2.4**		**2.3–4.3**
Lavandulol		0.8	0.3–0.5		0.4–1.5
**Terpinen-4-ol**	**2.3**	**2.7**			**3.4–5.3**
Cryptone + *p*-Cymen-8-ol	0.6				
Hexyl butyrate					0.3–0.5
α-Terpineol	0.9	1.2	2.1–2.6		1.5–1.6
γ-Terpineol					
**Linalyl acetate**	**23.6**	**26.2**	**29.1–32.3**	**17.8**	**18.6–34.2**
**Lavandulyl acetate**	2.2	2.3	2.4–2.7	**3.1**	1.5–2.6
Geranyl acetate		1.2			
(*E*)-β-Farnesene		1.1			
α-Santalene		0.2			
Germacrene D		1.1			
(*E*)-β-Caryophyllene			1.7–1.9	1.9	

ND—not determined, FF—fresh flowers, DF—dried flowers.

**Table 7 molecules-28-02943-t007:** Exemplary *L. × intermedia* ‘Abrial’ essential oil compositions from different countries.

Compound	France [72]	France [81]	Norway [61]	Spain [55]
	ND	ND	DF	DF
	Percentage (%)			
α-Pinene	0.9	0.4		
Camphene	0.6	0.3		
Octan-3-one		1.0		
Sabinene		0.1		
Oct-1-en-3-ol		0.3		
β-Pinene	0.9	0.3		
Myrcene	1.2	0.3		
Limonene	1.5	0.7		1.1
**1,8-Cineole**	**10.3**	**7.6**	**8.8**	**8.4**
(*Z*)-β-Ocimene	2.6	2.6		3.5
**(*E*)-β-Ocimene**	**4.2**	**3.0**		
Terpinolene		0.2		
**Linalool**	**19.6**	**35.0**	**31.1**	**41.9**
**Camphor**	**12.2**	**8.9**	**7.5**	**10.3**
*trans*-Linalool oxide		0.2		
*cis*-Linalool oxide		0.1		
**Borneol**	**3.7**	2.9		2.2
Lavandulol	0.6	0.6		0.8
Terpinen-4-ol	1.2			1.1
α-Terpineol	1.0	0.5		
**Linalyl acetate**	**18.6**	**27.0**	**17.2**	**22.0**
Lavandulyl acetate	2.6	1.0	2.9	1.7
Hexyl tiglate		0.3		
Neryl acetate		0.7		
Geranyl acetate	1.2	0.3		
(*E*)-β-Farnesene	1.2	0.3		
(*E*)-β-Caryophyllene		0.7	1.9	
α-Santalene		0.2		
Lavandulyl butyrate		0.2		
Germacrene D	1.2			
Caryophyllene oxide		0.3		

DF—dried flowers, ND—not determined.

**Table 8 molecules-28-02943-t008:** WHO and European pharmacopeia (Ph. Eur.) standards for a chromatographic profile of *Lavandula* sp. essential oils.

Compound	Content of Regulated Components [%]		
	Ph. Eur.	WHO	Ph. Eur.
	LA [82]	LA or LI [83]	LL [82]
Linalool	20–45	20–45	34–50
Linalyl acetate	25–47	25–46	<1.6
1,8-Cineole	<2.5	<2.5	16–39
Camphor	<1.2	<1.2	8–16
Limonene	<1	<1	0.5–3
Terpinen-4-ol	0.1–8	1.2–6.0	
α-Terpineol	<2	<2.0	0.2–2
Lavandulyl acetate	>0.2	>0.1	
Lavandulol	>0.1		
Octan-3-one	0.1–5	<2.5	
*trans*-α-Bisabolene			0.4–2.5

**Table 9 molecules-28-02943-t009:** Exemplary *L. × intermedia* hydrolate compositions (and their corresponding EOs) from different countries.

Plant Material Origin Reference	Australia [73]		Australia [73]		California [77]		Italy [76]		Spain [92]	Turkey [14]	
Material	EO	H	EO	H	EO	H	H	H	H	EO	H
Compound	Percentage (%)										
β-Pinene	1.3				0.9	0.1					
Myrcene					0.9	0.1	1.7	1.4			
β-Phellandrene	**7.9**										
Limonene					1.1	0.6					
**1,8-Cineole**	**15.3**	t	**11.6**	1.7	**7.4**	1.6	**25.4**	**28.9**	**5.0**	2.6	**9.8**
**(*Z*)-β-Ocimene**	**6.5**										
γ-Terpinene					1.6	0.1					
**Linalool**	**36.1**	**19.0**	**12.1**	**19.9**	**29.6**	**68.5**	**43.8**	**34.4**	**14.6**	**34.0**	**55.6**
**cis-Linalool oxide (f)**	0.1	2.8	**4.6**	t		0.8	0.1	t	**7.8**	0.4	**6.0**
**trans-Linalool oxide (f)**		**3.2**	**4.2**	t		0.6			**7.4**		
**Camphor**	0.7	2.4	**20.3**	**17.5**	**8.1**	0.8	**12.8**	**15.4**	**9.9**	**4.8**	**13.4**
**Borneol**	**0.7**	**5.2**	**6.0**	**31.8**	**2.5**	**1.4**	**4.3**	**4.0**	**9.3**	**4.2**	**13.5**
Lavandulol					0.5	1.4					
**Terpinen-4-ol**	**3.5**	**14.0**		2.3	1.9	**4.4**	**4.5**	2.7	0.2		
**Cryptone**		**7.1**		**7.5**	1.3	0.1					
(Z)-Hex-3-enyl butyrate			1.9								
**α-Terpineol**	1.0	**24.0**		**10.1**	1.6	**9.0**	1.8	2.2	**14.6**		
Nerol						1.7					
**Geraniol**					0.7	**5.3**		0.2	1.4	0.3	1.6
**Linalyl acetate**	**5.8**		**9.3**				2.1	0.4		**47.7**	t
Lavandulyl acetate	0.8		0.2				0.5	1.0			
Neryl acetate					0.6	0.1	0.2	0.4		2.4	t
Geranyl acetate					1.5	0.1	0.3	1.1			
(*E*)-β-Caryophyllene					1.8	0.1	0.2	1.3		1.0	t
(*E*)-β-Farnesene						0.2	0.2	1.4		0.5	t
Caryophyllene oxide			1.1				t	0.2		0.3	t

H—hydrolate, EO—essential oil, t—trace < 0.05%; f—furanoid form.

## Data Availability

Not applicable.

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
