# Peer review of "Lavandula × intermedia*—A Bastard Lavender or a Plant of Many Values? Part I. Biology and Chemical Composition of Lavandin"

_molecules, 2023, doi:10.3390/molecules28072943_

Round 1
Reviewer 1 Report
Dear Authors,
Despite all advantages of the manuscript (actuality, large interest to readers, high scientific level of the manuscript, etc.) some minor shortcomings have been observed and they have to be eliminated before publication.
In the Title, words: or, and, of ... have to be in small letters.
In the Introduction, you incorporated some abbreviations such LA, EOs, LI...but latter, through all the text you will prefer to use full name of “essential oils”, why?
Figure 1: Latin names of the plants have to be in italic.
Line 263: please check this line.
Line 313: a correct name is 3-carene
Table 3: a correct name is trans-Sabinene hydrate
Author Response
We would like to thank the Reviewer for the careful and thorough reading of this manuscript and for constructive suggestions, which help to improve the quality of our manuscript. We have tried to do our best to respond to any comments. As indicated below, we have considered all the general and specific comments provided by the Reviewer and have incorporated changes in the text to reflect them.
Point 1. In the Title, words: or, and, of ... have to be in small letters.
Corrected
Point 2. In the Introduction, you incorporated some abbreviations such LA, EOs, LI...but latter, through all the text you will prefer to use full name of “essential oils”, why?
That's a good point, and we agree that the inconsistency needed to be addressed. After reviewing the text, we introduced the abbreviations in many more places throughout the revised manuscript, using them interchangeably with the full or short names of the terms to avoid repetition and monotonous vocabulary.
Point 3. Figure 1: Latin names of the plants have to be in italic.
Corrected
Point 4. Line 263: please check this line.
Line 263 from Word manuscript: “Commercially L. x intermedia is grown to produce essential oils and for its horticultural importance – the most common cultivars are ‘Super’, ‘Groso’, and ‘Abrial’. Here we have corrected the typo in the sentence to ‘Grosso’.
Point 5. Line 313: a correct name is 3-carene
"Car-3-ene" is an allowed synonym for the compound. However, as suggested, we have corrected it to "3-carene" in the manuscript.
Point 6. Table 3: a correct name is trans-Sabinene hydrate
We have corrected trans-Sabinen hydrate to trans-Sabinene hydrate.

Reviewer 2 Report
The manuscript is very basic, it must add pharmacological activity of the essential oil.
In table 4 it mentions 11 countries, but in the tables of the report of the chemical composition of the essential oil it does not mention all of them, for example Romania. Please make a table of essential oil reports according to those reported in table 4.
Author Response
We would like to thank the Reviewer for the careful and thorough reading of this manuscript and for constructive suggestions, which help to improve the quality of our manuscript. We have tried to do our best to respond to any comments. As indicated below, we have considered all the general and specific comments provided by the Reviewer and have incorporated changes in the text to reflect them.
"The manuscript is very basic, it must add pharmacological activity of the essential oil."
Our article, titled "Lavandula x intermedia – A Bastard Lavender or a Plant of Many Values?" is divided into two parts. Part I, which the Reviewer has read, focuses on the biology and chemistry of lavandin, while Part II discusses the biological/pharmacological activity of essential oil and other lavandin preparations. We submitted both parts to the Molecules journal simultaneously and requested that the same reviewers assess both. Unfortunately, different reviewers were assigned to each part.
As an attachment to this reply, we are sending the complete manuscript of Part II, which has already been accepted for publication. It is awaiting the completion of the review process for Part I before publication. We kindly ask the Reviewer to see it in case there are any further doubts in this regard.
In Part II, we reviewed all original articles related to Lavandula x intermedia and its biological/pharmacological effects and included only studies with evidence-supported effects. Tables 1-3 in this manuscript provide a comprehensive summary of the current research in the field. Most studies are basic and performed on in vitro models. We were surprised by the lack of research on Lavandula x intermedia in the medical field, as Lavandula angustifolia is much more studied, including multiple clinical studies with Silexan. However, this is reflective of the current state of research in the field. The biological and application part of the article is quite lengthy, which is why we decided to split it into two parts (Part II alone has 37 pages).
Based on the reviewer's feedback, we acknowledge the need to reinforce that our manuscript is Part I of the article and add a reference to Part II for biological activities in the Introduction section. Therefore, we have added a reference to Part II for the biological activities (in the introduction and conclusion), so that readers are aware of the comprehensive nature of our work. We hope this improves the clarity of our manuscript and thank the Reviewer for this valuable feedback.
"In table 4 it mentions 11 countries, but in the tables of the report of the chemical composition of the essential oil it does not mention all of them, for example Romania. Please make a table of essential oil reports according to those reported in table 4."
Thank you for drawing our attention to this table. We have reviewed it and noticed that Romania and Turkey were in two separate lines, but they could be merged into one. We have corrected this and alphabetically ordered the lines. Additionally, instead of providing merged citations such as "Based on [2,6,58–65]," we have provided a reference for each record, which readers can use for further information. As the table was about oil yield we did not include the chromatographic profiles for the mentioned essential oils. We also want to note that many compositions of oil cited in this table were presented in other tables as well (such as Kara & Baydar 2013, Carrasco et al. 2016, Chakopoulou et al. 2003, Renaud et al. 2001), and therefore, we see no point in duplicating this information in the manuscript.
We hope this clarification meets the Reviewer's expectations. Thank you again for the valuable feedback.

Reviewer 3 Report
Ref.: molecules-2238616
Title: Lavandula x intermedia – A Bastard Lavender Or A Plant Of Many Values? Part I. Biology And Chemical Composition Of Lavandin
Authors: Pokajewicz et al.
Reviewer’s comments for author(s)
The authors provided an interesting manuscript that can be accepted after several corrections.
“Oil” and “essential oil” do not have the same meaning, so they should not be used as synonyms. “Oil” refers to vegetable oil, often obtained by expression of seeds, whereas “essential oil” refers to a product obtained either by a distillation procedure of any plant part, or expression only in the case of citrus epicarp. Th entire document should be checked to correct the presence of the word “oil” alone and substitute by “essential oil”.
European pharmacopeia should be included in the reference list.
Table 1. Please position the full table in one page. In the legend of this table, the scientific name should b in italics. The source of this classification should be cited.
References must be cited fully. For instance, check and complete the reference of the book of Marie Lis-Balchin, Lavender: the Genus Lavandula.
Table 2. Please consider changing the layout of the table in order that the titles “Plant habit”, “Inflorescence” and “Inflorescence” move to a new column on the left side. Likewise, for “shape”, “colour”, “stalk”, “spike” and others, like the example below:
|
|
|
Lavandula angustifolia |
Lavandula latifolia |
Lavandula x intermedia |
|
Leaves |
Shape |
narrow, linearlanceolate; |
linear-lanceolate to spathulate |
linear-lanceolate to spathulate. |
|
|
Color |
xxxx |
xxxx |
xxxx |
Table 2 should be full in one page.
Photos of Figure 2 should have the same hight. If possible, a scale should be added.
“Protective trichomes” are better known as covering trichomes.
The sentence “Essential oils accumulate in the subcuticular spaces and diffuse outwards when the cuticle is broken”, should be corrected. What accumulates in the subcuticular space is a secretion composed of different types of molecules. Only a part of this secretion is removed as an essential oil, since the distillation procedures extract what is volatile under the extraction conditions. For this reason, the sentence would be better reformulated for instance as “The secretion accumulates in the subcuticular spaces and diffuses outwards when the cuticle is broken”.
Figure 3 is of poor quality and should be removed or substituted by better images.
Table 3. Organize the table by alphabetic order of the name of the varieties. If possible, the table should b full in one page. If not, please repeat the header row at the top of each page used by the table.
Table 4. Countries should be arranged by alphabetic order. Each row should have its specific references. So, an additional column should be added in the right side with the corresponding references from ach country.
In the sentence “Other dominant terpenoids found in lavandin oil include camphor (2-33%, usually 5-10%)”, what does it mean the “usually 5-10%”? Similar question to the remaining values of the period of the same paragraph.
Figure 4 is not necessary.
Table 5. The header should have a defined criterion, applied also at all other similar tables. If the criterion is the country, then use alphabetic order of the names of the countries, from left to right. If the criterion is the plant part, then this row should be the first one, also arranged by alphabetic order. Maybe the line with the reference could be the last one?
All ISO publications are also protected by copyright. You may not lend, lease, reproduce, distribute or otherwise commercially exploit ISO publication(s). For this reason, you can compare the data (for instance just in terms of main components), but you cannot reproduce the complete data on the Table 6, 7 or 8.
Concerning the Lavandula x intermedia and L. angustifolia hydrolates, check Phytochem Rev (2022) 21:1661–1737 for additional references.
Table 9 caption as well as header should also be reformulated. The caption does not include the EOs, which are included in the table for comparison. Again, the header should be organized by alphabetic order of the country. To better highlight the differences between samples, maybe the threshold level should be higher, and include only compounds present at least once ≥3%.
When citing an author, the reference number should come following the authors name, and not at the end of the paragraph. For instance, instead of “Dobros et al. determined”, it should be “Dobros et al. [89] determined”.
The text can be improved editorially.
Author Response
We would like to thank the Reviewer for the careful and thorough reading of this manuscript and for constructive suggestions, which help to improve the quality of our manuscript. We have tried to do our best to respond to any comments. As indicated below, we have considered all the general and specific comments provided by the Reviewer and have incorporated changes in the text to reflect them.
Specific remarks:
“Oil” and “essential oil” do not have the same meaning, so they should not be used as synonyms. “Oil” refers to vegetable oil, often obtained by expression of seeds, whereas “essential oil” refers to a product obtained either by a distillation procedure of any plant part, or expression only in the case of citrus epicarp. Th entire document should be checked to correct the presence of the word “oil” alone and substitute by “essential oil”.
We appreciate the Reviewer’s insight into using the proper terminology and acknowledge that the most accurate term is "essential oil”. We have taken the comment into consideration in our revised manuscript.
European pharmacopeia should be included in the reference list.
Ph. Eur. was referenced in the submitted manuscript as reference 79. After correction, in the revised manuscript it is reference number 82.
Table 1. Please position the full table in one page. In the legend of this table, the scientific name should b in italics. The source of this classification should be cited.
We appreciate the feedback regarding the formatting of the Table and have made the necessary corrections to ensure it fits on a single page. Additionally, if our paper is accepted for publication, the final typesetting will be carried out in collaboration with the editorial team. Furthermore, we have made the necessary correction by italicizing the name of the plant in the table caption, as suggested by the reviewer.
References must be cited fully. For instance, check and complete the reference of the book of Marie Lis-Balchin, Lavender: the Genus Lavandula.
We have completed the reference in the revised manuscript.
„Lis-Balchin, M. Lavender: The genus Lavandula; Lis-Balchin, M., Ed.; 1st ed.; Taylor & Francis: London and New York, 2002; ISBN 0-415-28486-4.”
Table 2. Please consider changing the layout of the table in order that the titles “Plant habit”, “Inflorescence” and “Inflorescence” move to a new column on the left side. Likewise, for “shape”, “colour”, “stalk”, “spike” and others, like the example below:
The table was corrected according to suggestions.
Table 2 should be full in one page.
Corrected.
Photos of Figure 2 should have the same height. If possible, a scale should be added.
The figure was corrected according to the suggestions.
“Protective trichomes” are better known as covering trichomes.
Corrected.
The sentence “Essential oils accumulate in the subcuticular spaces and diffuse outwards when the cuticle is broken”, should be corrected. What accumulates in the subcuticular space is a secretion composed of different types of molecules. Only a part of this secretion is removed as an essential oil, since the distillation procedures extract what is volatile under the extraction conditions. For this reason, the sentence would be better reformulated for instance as “The secretion accumulates in the subcuticular spaces and diffuses outwards when the cuticle is broken”.
The sentence was corrected according to the Reviewer’s suggestion.
Figure 3 is of poor quality and should be removed or substituted by better images.
The figure was exchanged according to the Reviewer’s suggestion.
Table 3. Organize the table by alphabetic order of the name of the varieties. If possible, the table should b full in one page. If not, please repeat the header row at the top of each page used by the table.
We have taken this comment into consideration and made the necessary changes to organize the table in alphabetical order according to the name of the plant variety. The table header was repeated on each page as suggested.
Table 4. Countries should be arranged by alphabetic order. Each row should have its specific references. So, an additional column should be added in the right side with the corresponding references from ach country
We have reviewed Table 4 and identified that some countries had values in multiple columns. To ensure clarity and accuracy, we have added citations next to the values for these countries, indicating the sources of the data. We hope that this correction addresses any concerns and improves the overall quality of the manuscript.
In the sentence “Other dominant terpenoids found in lavandin oil include camphor (2-33%, usually 5-10%)”, what does it mean the “usually 5-10%”? Similar question to the remaining values of the period of the same paragraph.
Essential oils can vary greatly depending on the cultivar, origin, weather conditions, and other factors, so the percentage values reported in the literature can be quite wide-ranging. For example, according to Aprotosoaie et al. (2017), linalool and linalyl acetate can constitute anywhere from 2–48% of lavandin essential oil. Similarly, other components may also vary widely. However, we believe that presenting only such wide ranges without specifying the most frequent narrower range is less informative. Therefore, based on literature values provided by Aprotosoaie in the review and our former publication (Pokajewicz et al. 2022), we have narrowed down the ranges to the most frequent values.
The referred sentence was corrected to be more understandable.
- Aprotosoaie, A.C.; Gille, E.; Trifan, A.; Luca, V.S.; Miron, A. Essential oils of Lavandula genus: A systematic review of their chemistry. Phytochem. Rev. 2017, 16, 761–799.
- Pokajewicz, K.; Białoń, M.; Svydenko, L.; Hudz, N.; Balwierz, R.; Marciniak, D.; Wieczorek, P.P. Comparative Evaluation of the Essential Oil of the New Ukrainian Lavandula angustifolia and Lavandula x intermedia Cultivars Grown on the Same Plots. Molecules 2022, 27, doi:10.3390/molecules27072152.
Figure 4 is not necessary.
If the reviewer agrees, we would prefer to keep the Figure as we do not believe it negatively impacts the value of the manuscript.
Table 5. The header should have a defined criterion, applied also at all other similar tables. If the criterion is the country, then use alphabetic order of the names of the countries, from left to right. If the criterion is the plant part, then this row should be the first one, also arranged by alphabetic order. Maybe the line with the reference could be the last one?
Tables 5, 6, and 7 present exemplary compositions of lavandin essential oils. Table 5 pertains to the cultivar 'Super,' Table 6 to 'Grosso,' and Table 7 to 'Abrial.' The selection criterion was based on an example publication/country, and sometimes two different compositions from one country were presented. We have corrected the tables as per the reviewer's suggestion to arrange them in alphabetical order according to country names. We have also removed the column with the ISO standard, as suggested in one of the points below.
All ISO publications are also protected by copyright. You may not lend, lease, reproduce, distribute or otherwise commercially exploit ISO publication(s). For this reason, you can compare the data (for instance just in terms of main components), but you cannot reproduce the complete data on the Table 6, 7 or 8.
Thank you for bringing this important copyright issue to our attention. We have removed the column with ISO terpene specifications from all the tables.
Concerning the Lavandula x intermedia and L. angustifolia hydrolates, check Phytochem Rev (2022) 21:1661–1737 for additional references.
We would like to acknowledge that we have carefully reviewed the suggested publication, and we have found that most of the references published in Phytochem Rev were already included in our manuscript's Table 9. We would also like to note that in the revised manuscript we have cited the article the Reviewer has recommended.
Table 9 caption as well as header should also be reformulated. The caption does not include the EOs, which are included in the table for comparison. Again, the header should be organized by alphabetic order of the country. To better highlight the differences between samples, maybe the threshold level should be higher, and include only compounds present at least once ≥3%.
Based on Reviewer’s suggestion, we have reformulated the caption of the table to ensure greater clarity and precision. Additionally, we have organized the header of the table alphabetically to enhance readability.
Furthermore, we have adjusted the table to include major components, as per your recommendation, with a threshold of at least 1%. We would like to clarify that we have explored the possibility of increasing the threshold level to three percent but found no significant difference in capturing variations between the samples. We believe this may be attributed to differences in the method of volatile compound isolation. We would also like to emphasize the importance of including trace amounts of volatile compounds in the hydrolates composition analysis, as they can have a significant impact on the odor of the sample.
When citing an author, the reference number should come following the authors name, and not at the end of the paragraph. For instance, instead of “Dobros et al. determined”, it should be “Dobros et al. [89] determined”.
The text was reviewed and corrected accordingly.
The text can be improved editorially.
The whole manuscript was revised again and corrected.